# Patients with Persistent Polyclonal B-Cell Lymphocytosis Share the Symptomatic Criteria of Systemic Exertion Intolerance Disease

**DOI:** 10.3390/jcm10153374

**Published:** 2021-07-29

**Authors:** Romain Morizot, Jean-Dominique de Korwin, Pierre Feugier, Julien Broséus, Xavier Troussard, Jean-François Lesesve

**Affiliations:** 1Services d’Hématologie Clinique et Biologique, CHRU de Nancy, Université de Lorraine, F-54000 Nancy, France; R.MORIZOT@chru-nancy.fr (R.M.); p.feugier@chru-nancy.fr (P.F.); j.broseus@chru-nancy.fr (J.B.); 2Département de Médecine Interne et Immunologie Clinique, CHRU de Nancy, 54000 Nancy, France; jd.dekorwin@chru-nancy.fr; 3Laboratoire d’Hématologie, CHU de Caen, 14033 Caen, France; troussard-x@chu-caen.fr

**Keywords:** persistent polyclonal B-cell lymphocytosis, chronic fatigue syndrome, myalgic encephalomyelitis, systemic exertion intolerance disease, patient opinion

## Abstract

Introduction: Persistent polyclonal B-cell lymphocytosis (PPBL) is a rare and still poorly understood entity, with 90% of cases occurring in female smokers. Patients often appear tired and in pain, but the clinical symptoms remain imprecise. The main risk is the development of lymphoma in some cases. To better understand the characteristics of the fatigue associated with PPBL and study its relationship with systemic exertion intolerance disease (SEID), we analyzed the symptoms in a cohort of patients with PPBL included in the French national registry. Material and methods: An anonymous questionnaire following the recommendations of the Institute of Medicine/National Academy of Medicine for screening of the new SEID criteria was created in French and mailed to 50 patients. Results: Thirty-nine (78%) contacted patients responded. The studied population was mainly constituted of women (90%) with an average age of 50 (18–59) years. Smoking was a constant factor in all patients. A total of 28/39 (72%) respondents met the SEID symptoms criteria. Severe chronic fatigue for more than 6 months was noted in 36/39 cases (92%). Unrefreshing sleep, post-exertional malaise, cognitive impairment, and orthostatic intolerance were described in 30/39 (77%), 32/39 (82%), 28/39 (72%), and 27/39 (69%) cases, respectively. Pain (arthralgia, myalgia, headache) was present in 26/39 (67%) cases. The most prominent SEID symptoms were fatigue, followed by post-exercise discomfort and cognitive difficulties. The most disabling symptom was non-restorative sleep, followed by pain. An inflammatory and/or autoimmune context was noted in 13 patients (33%), and these comorbidities could have favored the deterioration of the general condition. Three patients also presented with fibromyalgia. However, 3 patients did not mention any complaints. Conclusion: This survey indicated that patients with PPBL most often initially presented with disabling chronic fatigue, chronic pain, and other symptoms suggestive of SEID but requiring more studies to confirm it. Education of medical staff about the symptoms of PPBL should be encouraged to better assess this peculiar condition.

## 1. Introduction

Persistent polyclonal B-cell lymphocytosis (PPBL) is a rare entity predominantly affecting middle-aged smoking women [1,2]. Less than 200 cases have been reported worldwide, mainly as isolated case reports or collective short series [3,4]. Moreover, as compared to frequent extensive biological investigations leading to the diagnosis, patients’ clinical complaints were not mentioned in most reports, wrongly suggesting that the patients were asymptomatic or poorly symptomatic. Weariness or fatigue have only occasionally been mentioned in the literature [1,5,6]. French patients with PPBL are regularly monitored, as most of them are enrolled in a national registry comprising more than 70 patients, offering the largest series of PPBL cases worldwide. Moreover, most of them complain about fatigue, widespread musculoskeletal pains, rheumatic symptoms, or migraines. Some patients also presented with fibromyalgia [7]. Thus, we aimed to investigate the presenting symptoms in patients with PPBL more accurately by administering an anonymous written questionnaire to the patients included in the French registry.

Empirically, it seemed that some PPBL symptoms were comparable to those described in myalgic encephalomyelitis/chronic fatigue syndrome (ME/CFS). ME/CFS is a poorly understood multi-system disorder characterized by lasting symptoms for at least six months, including severe incapacitating fatigue, cognitive dysfunction, sleep disturbances, muscle pain, and post-exertional malaise (PEM), all of these resulting in substantial reduction in functional activity and quality of life [8,9,10]. Recently, a new case definition for ME/CFS and a new name, “systemic exertion intolerance disease” (SEID), have been proposed by the Institute of Medicine, now called the National Academy of Medicine (IOM/NAM, 2015) [11], exclusively based on the following clinical examination criteria: substantial reduction or impairment of occupational, educational, social, or personal activities for more than six months, accompanied by profound fatigue not alleviated by rest; PEM; and unrefreshing sleep. All three criteria are required for SEID diagnosis. Cognitive impairment and/or orthostatic intolerance (OI) could also be present. A specific questionnaire following the recommendations of the IOM/NAM (“tool kit”) for the screening of the new SEID criteria was created in French [12]. In this survey, we sent this questionnaire and additional items to patients with PPBL. Here, we report the evaluation of the general feelings, SEID criteria, and perception of medical staff’s attitude among patients with PPBL, as reported by the patients in response to the questionnaire.

## 2. Material and Methods

### 2.1. Population

All patients were previously included in the French registry for PPBL (promoted by the Research and Innovation Direction from Nancy University Hospital (#12.114, 3 March 2012); national ethics agreement (#DR-210, 14 May 2012)). At the time of the study, the registry included the data of 79 patients, 50 among them being exhaustively investigated and regularly followed up (clinical evaluations, full blood count, lymphocyte immunophenotyping, HLA investigation, serum immunoglobulins measurement, genetics, etc.).

### 2.2. Questionnaire Structure and Validation

First, a letter describing the project, questionnaire, and aims of this survey were sent to the 50 patients. The questionnaire was proposed to these patients to investigate their symptoms and feelings about their health, focusing particularly on fatigue, pain, and evaluating the SEID criteria. The respondents who voluntarily completed the survey remained anonymous for subsequent data processing. The questionnaire was built by the ASFC Scientific Committee, composed of ME/CFS French experts, and previously tested by five randomly chosen members of ASFC [12]. Their assessment was that the questions, translated by a professional, were clear and understandable, and truly corresponded to their feelings. SEID criteria were evaluated by questions derived from the practical recommendations proposed by the IOM/NAM “tool kit” [11], giving examples of terms ME/CFS patients commonly use to describe their symptoms and potential questions that can alert clinicians to the diagnosis. All patients were screened for exclusionary conditions. Patients had to respond to each item to progress to the next question. The items concerned (i) demographics; (ii) SEID criteria: disabling fatigue (duration, sudden or progressive onset, history of an infection or without a trigger, delay between fatigue onset and PPBL diagnosis, severity, onset after exercise or mild exertion), unrefreshing sleep, PEM and exertional exhaustion after increased physical effort, cognitive impairment (memory, concentrating), and OI; (iii) pains (musculoskeletal, headache); (iv) other symptoms and signs (lymph nodes, sore throat, sweats); and (v) opinions regarding medical staff and their attitude. To evaluate severity, the subjects had to declare if a symptom was present (more or less than 50% of the time) or not and rate the main symptoms according to the impairment degree. An item was considered positive if at least 1 symptom was present for more than 50% of the time. Duration was evaluated over the past 6 months to more than 3 years, and the severity of symptoms was addressed using an anchored ordinal scale of 0 (no symptoms), 1 (moderate), and 2 (severe). Apart from the IOM questionnaire, free text comments allowed respondents to list their impressions and to express their opinions. 

### 2.3. SEID Criteria

To be classified as an SEID patient according to the diagnostic criteria for ME/CFS proposed by the IOM/NAM [11], respondents had to meet the three mandatory criteria (disabling fatigue, PEM, unrefreshing sleep) and at least one of the two facultative criteria (cognitive impairment or OI). The SEID criterion was adopted in the case of a positive answer to one or more symptom-related questions, and if the symptom was present at least 50% of the time: disabling fatigue (3 questions), PEM (4 questions), unrefreshing sleep (3 questions), cognitive impairment (9 questions), and OI (4 questions).

### 2.4. Data Processing

Data collection was stopped after 8 months (one recall of non-respondents after 6 months). Responses were copied to Microsoft Excel software. Data were reported as numbers and/or percentages for categorical variables. Free comments were reviewed, collated, and codified according to the common themes. Number of comments per theme were computed, including the frequencies of the most common symptoms.

## 3. Results 

### 3.1. Characteristics of Subjects 

Thirty-nine subjects among 50 (78%) returned the questionnaire on time. All but two questionnaires were 100% complete. Demographic data and PPBL characteristics are described in Table 1. The study population mainly included women (90%), and the most prevalent age was 45 to 55 years.

### 3.2. SEID Criteria

#### 3.2.1. SEID Diagnosis

Among 39 evaluable patients, 28 (72%) fulfilled the SEID symptoms criteria, with the presence of the 3 mandatory criteria (disabling chronic fatigue, unrefreshing sleep, PEM) and at least 1 of the 2 optional criteria (cognitive disorders, OI) (Figure 1). However, 3 patients did not mention any symptoms and remained fully asymptomatic during follow-up.

#### 3.2.2. Fatigue 

The delay between the fatigue onset and the diagnosis was substantial, with an average of 4 years, PPBL diagnosis being always made after fatigue occurrence. All but 5 subjects declared that their fatigue had lasted for more than 6 months. Moreover, 25 subjects had even suffered from fatigue for more than 3 years. Only 9 individuals reported what they considered as a trigger, which was infectious for 3 of them, including viral hepatitis, Lyme disease, and repeated cystitis. Fatigue onset was almost unanimously progressive, with only 5 patients describing a sudden onset. (Table 2).

General practitioners were in the front line for investigations (for 29 patients) and specialists (internists, rheumatologists, and hematologists) conducted investigations for 10 patients. Two or more clinicians were consulted for all patients. None but 2 of the respondents had visited a physician with a previously known diagnosis of CFS. The patients mostly described themselves as exhausted (“discharged battery”) and mentioned an aftereffect on their memory. They insisted on the difficulty in practicing physical exercises and increases in fatigue after mild exertion. However, the fatigue was considered as banal in 26 of the cases. Consequently, fatigue symptoms were fully taken into account by the physicians mostly after numerous visits and after confirming PPBL diagnosis for 6 patients. Finally, fatigue was considered as disabling for 36/39 patients (92%).

#### 3.2.3. Unrefreshing Sleep

More than half the patients declared that they felt “like they never slept” and “not good in the morning even after a normal night’s sleep”. Unrefreshing sleep (i.e., feeling tired upon waking or in the same state as before bedtime) was among the most common symptoms reported, and only 6 (15%) patients failed to report some type of sleep dysfunction. More than a third of the patients reported sleep disturbances (nonrestorative sleep or insomnia), and morning stiffness was reported more than half the time, in addition to fatigue. Finally, sleep was considered unrefreshing sleep for 30/39 (77%) responders.

#### 3.2.4. PEM and Exertional Exhaustion after Increased Physical Effort

Nearly half the patients felt “crashed, relapsed, or collapsed” more than half the time. In addition, 75% of the respondents felt significantly more “mentally tired after the slightest effort”. Thirty-two (82%) patients highlighted severe fatigue after mild exertion. Finally, the validity of a PEM was assessed for 32/39 (82%) responders.

#### 3.2.5. Cognitive Disorders (Memory, Concentration)

The patients felt “confused” and “disoriented” for only around 50% of the responses. Those who felt confused always preferentially rated “less than half the time” for all items. The main disorders were linked to an immediate memory impairment (concentration and thinking). Nevertheless, 28/39 (72%) responders complained of a cognitive impairment.

#### 3.2.6. Orthostatic Intolerance 

Only a minority of patients described major vertigo or dizziness. Five felt empty-minded after staying up for a while, 27 did not feel good, 14 felt dizzy, and the symptoms improved after going to bed for 11 patients. However, while only 4 patients answered “yes” to 3 out of the 4 questions, 27 answered “yes” to 1 item. According to the IOM criteria, these 27/39 patients’ (69%) results were consistent with having an OI.

#### 3.2.7. Asymptomatic or Poorly Symptomatic Patients

Eleven patients of the 39 (28%) did not meet the SEID symptoms criteria. Nevertheless, 6 among them described profound chronic fatigue not alleviated by rest, but the other 2 mandatory symptoms were missing (PEM in 3 and unrefreshing sleep in 3 patients). More intriguingly, 5 patients did not complain of debilitating fatigue (quoted as “mild”; duration: less than 6 months for 4 and less than 1 year for 1 patient), PEM, unrefreshing sleep, cognitive impairment, and OI. Careful interviewing of these 5 patients confirmed these findings in 3 patients (female patients, one with spleen enlargement), but highlighted high resilience levels for a male patient with a child suffering from severe illness (allografted juvenile myelomonocytic leukemia) and a female patient with primary biliary cholangitis, both denying their own general status alteration.

#### 3.2.8. Relation between PPBL Information and Symptoms for SEID

We observed that the 6 patients with less severe SEID (mild fatigue, few if any cognitive impairments, preserved sleep) had a regular PPBL profile: no splenomegaly, usual immunophenotype and karyotype, no clinical nor biological autoimmunity. Conversely, more severe SEID patients were mostly painful, have autoimmune manifestations and more abnormal karyotypes. When focusing on the single fatigue item, out of the 19 patients with highly disabling fatigue, 14 had antinuclear antibodies and 3 had clinical autoimmune disease. Though pains are not included in SEID criteria, patients reporting pain had a more severe SEID and positive biological investigations for autoimmunity. In our series, a connection between more severe SEID and splenomegaly occurrence, autoimmune manifestations, and pains might exist. 

All patients with comorbidities ranked their fatigue feelings higher. SEID criteria were more predominantly positive and pains more present for these patients. By contrast, no influences on cognitive manifestations or PEM were noticed.

### 3.3. Other Survey Items

#### 3.3.1. Pain

Headache and joint or muscle pains appeared as a quite frequent complaint of the patients at the first visit, often appearing as unspecific musculoskeletal pains and arthralgia. The survey showed that 26/39 (67%) respondents had uncomfortable pains. These pains were often intermittent, concerning the back, arms, and legs, and mostly in the joint. More than half of the patients suffered from headaches. Some patients also described diffuse erratic pains. Three women had a previously assessed diagnosis for fibromyalgia.

#### 3.3.2. Other Symptoms

Twenty-three subjects declared suffering from unusual sweats (59%) and 5 from sore throat. Eight patients had swollen lymph nodes (cervical localization, 2) or splenomegaly (6). Almost all patients declared psychosomatic complaints and almost all felt depressed and in a sad mood. There was no correlation between severity of fatigue and the number of complaints.

#### 3.3.3. Associated Inflammatory or Autoimmune Pathologies

Two patients presented with psoriasis and 12 (31%) presented with autoimmune disease: 4 patients with an inflammatory rheumatologic disease (2 rheumatoid arthritis, 2 ankylosing spondylitis), 4 with auto-immune thyroiditis, 1 with type 1 diabetes, 1 with primary biliary cholangitis, 1 with Crohn’s disease, and 1 with Sjögren’s syndrome. Thirteen (34%) patients had positive laboratory test for dysimmunity.

Six patients had a complex medical situation due to their autoimmune disorders (4) or breast cancer (2) that could explain a general alteration of their status.

#### 3.3.4. Opinions Regarding Medical Staff and Their Attitude

All patients had a primary care physician but claimed to have been diagnosed with PPBL only after visiting a specialist. Most respondents reported that the general practitioners were not knowledgeable about PPBL and were not truly listening to their current symptoms. Twelve subjects did not go to a physician for more than a year, stating “nothing could be done” or “they would not be considered seriously”. A few patients resigned, asserting that “they know that there was nothing medicine could do for them”.

### 3.4. Symptoms Ranking 

Chronic fatigue was clearly rated as the most disabling factor, followed by PEM and cognitive impairment. Unrefreshing sleep was identified as the most disturbing factor for 77% of the subjects, followed by pain (67%). Limitation of activities and failure to concentrate were often written in free comments.

## 4. Discussion

Due to their poor description in the previous literature, we investigated clinical symptoms of PPBL using the resource of the largest cohort of patients who have been thoroughly investigated through laboratory tests. PPBL is a rare condition that can present with a myriad of erratic somatic symptoms. As found in this series, the standard history highlighted that female were the most affected, and events were often related to chronic smoking habits. Laboratory investigations currently remain the cornerstone for assessing the diagnosis, including peripheral blood examination typically demonstrating an isolated moderate, chronic, and absolute lymphocytosis, with scarce atypical binucleated lymphocytes making the diagnosis often difficult [13]. High polyclonal serum IgM levels is the rule. Crucially, the B-cells appear polyclonal, with both kappa and lambda light-chain expression and no clonal rearrangement of immunoglobulin heavy-chain genes. The cytogenetic profile includes an additional isochromosome for the long arm of chromosome 3, +i(3)(q10) [14,15], and *MECOM* gene amplification [16]. HLA typing disclosed the presence of at least one DR7 antigen in 70% of the patients. Though PPBL remains a benign disease, approximately 3% of the patients are at risk of lymphoma transformation, justifying a precise diagnosis and follow-up [17]. This emphasizes the need for a more practical approach after the initial visit of the patient. 

As 39 out of 50 thoroughly assessed PPBL patients responded to the questionnaire, we assume this panel is representative of the PPBL population. This survey highlighted that nearly three-quarters of the patients with PPBL met the SEID criteria. Unnervingly, these patients often initially present with a complex list of chronic symptoms, including disabling chronic fatigue, PEM, and widespread pains that commonly remain questionable for the physician. By contrast, SEID has been deeply evaluated and was convincingly described as a debilitating disease characterized by fatigue, PEM, cognitive dysfunction, and sleep disturbances, as recurrently found in the PPBL subjects in our study. Contrasting with PPBL, SEID is not that rare, with a prevalence rate varying up to 2.5% in the US [18]. Our basic approach remains difficult and open to criticism. Our opinion results from the analysis of the responses to a questionnaire, including the five SEID criteria because of their simplicity, and the availability of the IOM/NAM items set, previously adapted and translated into French [8,11,12]. Our pilot-written survey primarily aimed to determine the common presentation of patients with PPBL, score the severity of core ME/CFS symptoms, identify reasons for going to the physician, and grade attitudes and impressions of care. Practically, we have operationalized the IOM/NAM criteria as an anonymous questionnaire that could quickly provide guidance about the main complaints. This instrument was used to assess the SEID criteria, pains, other symptoms, and opinions regarding medical staff in the PPBL population. Additional free text comments helped to describe attitudes of the patients with PPBL regarding their condition and interactions with physicians. 

All but two patients with PPBL suffered from a progressive long-lasting fatigue. Within unexplained chronic fatigue, ME/CFS is defined by a disabling long-lasting fatigue, without specific organic or psychiatric diseases, associated with different criteria, including physical or cognitive manifestations. In addition, the signs had to be present on at least 50% of the days in the past 6 months, chronicity being known as a key feature of the CFS diagnosis [19]. In our study, most of the patients with PPBL met these criteria. As for PPBL, ME/CFS affects more women, with a sex ratio of 4:1 [10,11,16]. However, fatigue is a common complaint, and linking it to ME/CFS remains challenging. Other noteworthy diagnostic criteria, such as fibromyalgia (three female patients), are not exclusionary criteria for SEID anymore, but are considered as comorbidities by the IOM/NAM [11]. Overlapping syndromes are usually considered in other classifications [8,10]. We insist on the necessity of considering this disabling fatigue, and even, at the first visit, of suspecting PPBL as one of the possible diagnoses. 

PEM was assessed in 82% of all patients with PPBL. PEM is a crucial finding, and the evidence-based analysis of the IOM/NAM precisely renamed SEID as ME/CFS to emphasize the disability and importance of PEM for diagnosis. PEM is a necessary criterion in the most recent ME/SEID classifications [10,11]. PEM is an exacerbation of ME/CFS symptoms that occurs after physical or cognitive exertion and leads to a reduction in functional ability [20]. As observed in our study population, this symptom remains extremely difficult to assess. The way PEM is defined can affect how patients interpret the concept of PEM and whether they endorse it. PEM was recently shown to be composed of two empirically different experiences, including generalized fatigue and muscle-specific fatigue [21], both of which affect patients with PPBL. Respondents clearly indicated that their symptoms worsened with exertion. Commonly endorsed symptoms were reduced functional capacity and problem in logical flow if they walked a long distance. Cognitive disorders (memory disturbance, focusing) were mentioned less by the respondents (72%). Based on our results, patients with PPBL only felt partially “confused” and “disoriented”. Our survey also indicated the importance of OI (69%) as one of the causes for a visit to a physician. However, cognitive disorders or OI, although frequent in patients with ME/CFS, are not mandatory criteria for defining SEID [11]. Of note, three patients suffered from fibromyalgia, and fibromyalgia associated with CFS seemed to exacerbate PEM symptoms [20]. This could be because fibromyalgia and ME/CFS share common symptoms, such as chronic fatigue and muscle pain, suffering, and disability. 

Eleven (28%) patients of the cohort could not strictly be classified as having SEID, representing a substantial proportion of the cohort. Nevertheless, six presented with chronic fatigue, and the remaining five revisited their symptoms because of serious health concerns or family issues, supported by high resilience levels. It might be considered that these 11 patients, though strictly not fulfilling the criteria for SEID, were close to this condition. On the other hand, we had to consider rare patients that presented without any symptoms, some of them even remaining fully asymptomatic during follow-up. All patients without SEID or with less severe SEID had a regular PPBL profile, including the usual biological and karyotypic anomalies. Conversely, we felt that a link between more severe SEID and splenomegaly occurrence, autoimmunity, and pains might exist. However, as the answers to the survey were not uniformly scaled by a physician, this point remains to be prospectively investigated. As expected, all patients with comorbidities ranked their fatigue feelings higher, more predominantly quoted SEID criteria, and also reported more pain.

Concerning the SEID diagnosis, 28/39 (72%) of the PPBL patients fulfilled the symptoms criteria. In a previous study using the same questionnaire to evaluate SEID criteria, we found that 84% of subjects self-reporting ME/CFS diagnosis mainly based on Fukuda’s criteria [9] met the SEID criteria [12]. Other studies on SEID criteria have reported comparable results classifying a similar percentage of subjects among ME/CFS patients (72–88%) [22,23]. Nevertheless, the diagnostic value of SEID criteria remains controversial, being considered less specific than ME criteria [22,23,24].

A limitation comes from the current debate on SEID investigation. While the IOM do recommend the development of a toolkit appropriate for screening and diagnosing patients with ME/CFS, it does not actually detail the contents of this toolkit. As part of this toolkit, the IOM even feel that the development of clinical questionnaire or history tools that are valid across populations of patients should be an urgent priority [11]. Rigorous recommendations when applying SEID diagnostic criteria were investigated by Asprusten et al. (2018) with an adolescent CFS cohort [25]. This study used variables from a total of eight validated questionnaires to operationalize the SEID criteria, and then used baseline data to decide whether a patient fulfilled those criteria or not. Two CFS subgroups (SEID vs. non-SEID) were compared across baseline characteristics, as well as a wide range of cardiovascular, inflammatory, infectious, neuroendocrine, and cognitive variables. We did not go so far in our study, and this could be seen as a weakness for SEID specialists, but our main aim was to sensitize hematologists to the sad feeling of a majority of PPBL patients. In particular, the Aspruten study suggests that the SEID criteria tend to select patients with depressive symptoms. Here, we recognize that skepticism about the role of psychological factors in ME/CFS and high rates of depression in patients with PPBL can lead to misdiagnosis of ME/CFS as a psychiatric disorder. As we did not rigorously employ controls for depression, this might be discussed as a potentially confounding factor.

Many patients with PPBL had negative opinions regarding medical staff and their attitude, with medical care being sometimes compared to an obstacle course, often with delayed diagnosis. However, PPBL symptoms remain complex and difficult to explain in routine clinical practice. Skepticism about the role of psychological factors in ME/CFS and high rates of depression in patients with PPBL can lead to misdiagnosis of ME/CFS as a psychiatric disorder [26]. Moreover, in some instances, general practitioners might have negative attitude towards patients with poorly understood conditions, such as PPBL or ME/CFS. Our study highlighted these patients as a homogeneous group with common but medically unexplained symptoms. From the first visit, most patients deeply felt a barrier to care resulting from the lack of knowledge of PPBL symptoms and diagnosis of physicians. The feeling of being dismissed and not considered seriously was widespread among the patients. A few of them were even told that their symptoms were psychologically driven. When diagnosed by a specialist, some patients considered general physicians to be unknowledgeable about ME/CFS and PPBL. These factors make it imperative to understand the presentation of PPBL and educate medical staff about PPBL symptoms. Education of physicians about the modern concepts of ME/CFS is necessary to improve satisfaction among the patients and medical staff. Our survey also highlights the need for guidance in the diagnosis of PPBL. Since some SEID criteria could be more precisely investigated by appropriate questions focusing on exploring different components of the condition, we believe that a self-reported questionnaire such as ours might be useful as a screening instrument to check the patient’s complaints.

Though a written questionnaire gave the opportunity to ask many questions, it is possible that the length of the survey made it difficult to complete for some participants. The main limitation of this study is the difficulty in accurately attributing the symptoms of ME/CFS/SEID to their related disorders because of lack of clinical evaluation of the disease, the diagnosis being solely based on a questionnaire. In the absence of a specific questionnaire for the identification of SEID criteria, we created an original questionnaire, based on the recommendations of the IOM/NAM for the identification in patients of symptoms characteristic of SEID. The questions were formulated using the usual expressions of patients identified by the IOM/NAM [11] and validated by a preliminary test with members of the French ME/CFS patients’ association (ASFC). Further symptom surveys comparing SEID criteria and existing ME/CFS case definitions operationalized the case definition criteria using the DePaul Symptom Questionnaire symptom scale (DSQ) to obtain most symptoms covered by the different classifications [22,26]. However, DSQ has not already been validated in a French framework. A second major limitation was the coexistence of PPBL with other diseases also prone to induce chronic fatigue. Indeed, to establish SEID diagnosis, clinicians need to exclude other causes of the main symptoms required for the classification of patients [11]. However, in this study, a specialist conducted careful examination of all these patients included in the database. Finally, six patients presented with comorbidities that could explain a general deterioration in their condition. Nevertheless, even after exclusion of these patients, a significant number of patients (17, 53%) would have remained with a possible diagnosis of SEID.

To conclude, this study identified key complaints often reported by patients with PPBL. Nearly three-quarters of the subjects met the SEID criteria characterized by an original questionnaire. Most patients experienced chronic fatigue with a substantial impairment, unrefreshing sleep, PEM, cognitive disorders, and OI that always preceded biological PPBL diagnosis. These symptoms and widespread pains were the most severe and persistent problems in the previous six months, overlapping with the SEID criteria since initial presentation. It would be interesting to confirm these results with a similar survey in a larger worldwide cohort. To clarify the controversial relationship between these two entities, we also plan to conduct the reverse screening for binucleated lymphocytes in patients with ME/CFS according to the SEID criteria.

## Figures and Tables

**Figure 1 jcm-10-03374-f001:**
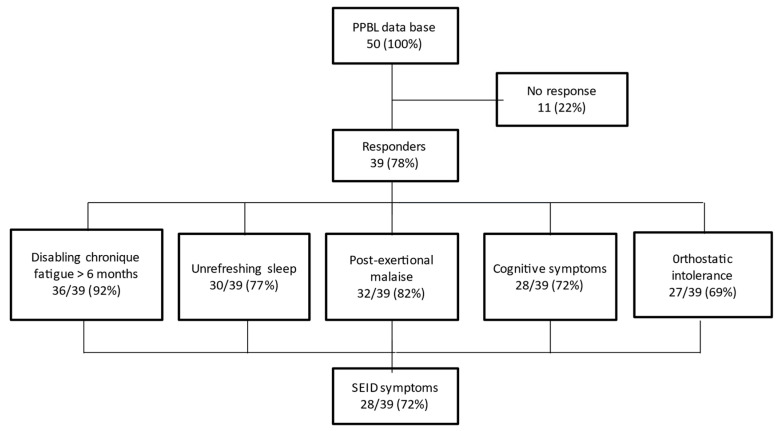
Summary of the survey highlighting the systemic exertion intolerance disease (SEID) criteria.

**Table 1 jcm-10-03374-t001:** Population data (*n* = 39).

Age (years)—Mean (min–max)	50 (18–59)
Sex (M/F)	4 (10%)/35 (90%)
Tobacco consumption	39 (100%)
Clinical presentation	
Lymph nodes	2 (5%)
Splenomegaly	6 (15%)
Auto-immunity (%)	12 (31%)
Full Blood Count—Mean (min–max)	
White blood cells (109/L)	12 (7.22–17)
Hemoglobin (g/dL)	13.9 (10–17.8)
Platelets (10^9^/L)	263 (68–540)
Lymphocytosis (10^9^/L)	5.55 (2.17–10.56)
Binucleated lymphocytes (patients; % of lymphocytes)	39; 5 (1–13)
IgM (g/L)—Mean (min–max)	6.3 (0.83–14.7)
HLA DR7 positive	21 (66%)
CD19% (immunophenotype)—mean (min–max)	50 (12–92)
Polyclonality (%)	100
Iso3q10 positive (karyotype or FISH)	27 (84%)
IgH gene rearrangement	6
Secondary neoplasia	breast cancer: 2
	Non-Hodgkin’s Lymphoma: 1

**Table 2 jcm-10-03374-t002:** Items explored by the survey.

**Chronic Fatigue**		**36/39**
Duration		
<6 months		5 (13%)
>1 year		2 (5%)
>2 years		4 (10%)
>3 years		25 (64%)
Onset		
Progressive onset		31 (86%)
Brutal onset		5 (14%)
Patients’ supposed trigger		
After polyarthritis		2
After spondylarthrtis		2
After neoplasia (osteosarcoma, breast)		2
After infection (cystitis, HVC)		3
Experienced as banal		26 (72%)
Experienced as alarming		10 (28%)
Patients’ feelings		
“I feel like I have a flu”		10 (28%)
“I feel like an exhausted battery”		24 (67%)
“I require more effort to think”		22 (61%)
Fatigue symptoms taken into account by the physician		
At first visit		2 (5%)
After PPBL diagnosis		6 (17%)
After numerous visits		24 (67%)
Never		4 (11%)
**Sleeping**		**30/39**
“I feel tired as if I have not slept”	less than half the time	10 (26%)
	more than half the time	23 (59%)
	never	6 (15%)
“Though I slept as usual, I woke up tired”	less than half the time	12 (31%)
	more than half the time	20 (51%)
	never	7 (18%)
“I can never feel rested”	less than half the time	13 (33%)
	more than half the time	17 (44%)
	never	9 (23%)
**After sustained physical or intellectual activity, or prolonged standing**		**32/39**
“I feel crashed, relapsed or collapsed”	less than half the time	10 (26%)
	more than half the time	20 (51%)
	never	9 (23%)
“I feel mentally worn out after light effort”	less than half the time	12 (31%)
	more than half the time	17 (44%)
	never	10 (26%)
“I feel physically emptied after moderate exertion”	less than half the time	12 (31%)
	more than half the time	20 (51%)
	never	7 (18%)
“The more effort my activities require me to put, the more I pay the price in return”	less than half the time	6 (15%)
	more than half the time	26 (67%)
	never	7 (18%)
**Intellectually weak to perform some activities (driving, reading, movie watching, computer working, discuss, or debate)**		**28/39**
“I feel foggy”	less than the half time	13 (33%)
	more than the half time	8 (21%)
	never	18 (46%)
“I feel confused”	less than the half time	16 (41%)
	more than the half time	5 (13%)
	never	18 (46%)
“I feel disorientated”	less than half the time	11 (28%)
	more than half the time	6 (16%)
	never	22 (56%)
“I have trouble in concentrating”	less than half the time	15 (38%)
	more than half the time	13 (34%)
	never	11 (28%)
“I am unable to process information correctly”	less than the half time	13 (33%)
	more than half the time	10 (26%)
	never	16 (41%)
“I find it difficult to find the right words”	less than half the time	19 (49%)
	more than half the time	6 (15%)
	never	14 (36%)
“I am not able to engage in several parallel activities”	less than half the time	19 (49%)
	more than half the time	7 (18%)
	never	13 (33%)
“I find it difficult to make choices and decisions”	less than half the time	15 (38%)
	more than half the time	7 (18%)
	never	17 (44%)
“I have absences and blackouts”	less than half the time	19 (49%)
	more than half the time	7 (18%)
	never	13 (33%)
**“When I stand up and stay up for a while”**		
“I feel empty-minded”	yes	5 (13%)
	no	34 (87%)
“I don’t feel good”	yes	27 (69%)
	no	12 (31%)
“I feel dizzy”	yes	14 (36%)
	no	25 (64%)
“I feel better when going to bed and lifting my legs”	yes	11 (28%)
	no	28 (72%)
**Pains**		**26 (67%)**
diffuse		18 (46%)
joints		26 (67%)
muscles		16 (41%)
arms and legs		21 (54%)
back		24 (62%)
headache		21 (54%)
intermittent		18 (46%)
permanent		9 (23%)
**Other symptoms**		
erratic pains		26 (67%)
throat irritation		5 (13%)
chills		20 (51%)
sweats		23 (59%)

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
