# Peer review of "Patients with Persistent Polyclonal B-Cell Lymphocytosis Share the Symptomatic Criteria of Systemic Exertion Intolerance Disease"

_jcm, 2021, doi:10.3390/jcm10153374_

Round 1

Reviewer 1 Report

My original recommendation was ”Reject (article has serious flaws, additional experiments needed, research not conducted correctly)”. I apologize if my comments were taken to suggest that expanding the rationale for conducting the study would be sufficient to warrant acceptance for publication.

Even if the value of applying SEID criteria to the evaluation of PPLB was accepted, I do not believe the study design and instruments used are fit for purpose. The article is misleading in implying that the questionnaire used was developed “following the recommendations of the IOM/NAM (“tool kit”) for the screening of the new SEID criteria”. While the IOM do recommend the development of “a toolkit appropriate for screening and diagnosing patients with ME/CFS”, they do not actually detail the contents of this “toolkit”.

As part of this toolkit the IOM feel that “the development of clinical questionnaire or history tools that are valid across populations of patients should be an urgent priority”. Toward this end Appendix D of their report comprises “Questionnaires and Tools That May Be Useful for Assessing ME/CFS Symptoms” with the footnote that “Further evidence of utility is needed”.

In my original review I mentioned a study by Asprusten et al (2018) which follows the IOM recommendations when applying SEID diagnostic criteria to an adolescent CFS cohort. This study used variables from a total of 8 validated questionnaires to operationalise the SEID criteria, and then used baseline data to decide whether a patient fulfilled those criteria or not. Two CFS subgroups (SEID vs Non SEID) were compared across baseline characteristics, as well as a wide range of cardiovascular, inflammatory, infectious, neuroendocrine and cognitive variables. I find no evidence to show that this present research or the original Campagne (2016) study employed a similar level of scientific rigor when developing their own SEID questionnaire.

Aspruten et als. findings question the discriminant and prognostic validity of the SEID diagnostic criteria in adolescent CFS. In their response to my comments, the present authors acknowledge “the controversy arising from the IOM/NAM's proposals to better define ME/CFS under the SEID label”. However, they fail to discuss the implications that a lack of discriminant and prognostic validity of the SEID diagnostic criteria might have for their use in both research and clinical practice.

The Aspruten study suggests that the SEID criteria tend to select patients with depressive symptoms. The present study does recognize that “skepticism about the role of psychological factors in ME/CFS and high rates of depression in patients with PPBL can lead to misdiagnosis of ME/CFS as a psychiatric disorder”. However, there were no controls for depression employed in the present study, nor is it discussed as a potentially confounding factor.

I was not aware of PPLB before reading this paper and thank the authors for drawing my attention to this rare condition. There is certainly potential benefit from further exploration of a possible link with ME/CFS. Is it a trigger for ME/CFS, a comorbidity, or a condition with symptoms that resemble ME/CFS? The possible link with EBV is certainly interesting and, of course, other possible viral etiologies such as COVID 19. PPLB should likely be considered for differential diagnosis when evaluating patients for possible ME/CFS. However, I do not believe this present study adds any clarity to the current situation regarding diagnosis and/or management of either ME/CFS or PPLB.

Author Response

My original recommendation was ”Reject (article has serious flaws, additional experiments needed, research not conducted correctly)”. I apologize if my comments were taken to suggest that expanding the rationale for conducting the study would be sufficient to warrant acceptance for publication.

Even if the value of applying SEID criteria to the evaluation of PPLB was accepted, I do not believe the study design and instruments used are fit for purpose. The article is misleading in implying that the questionnaire used was developed “following the recommendations of the IOM/NAM (“tool kit”) for the screening of the new SEID criteria”. While the IOM do recommend the development of “a toolkit appropriate for screening and diagnosing patients with ME/CFS”, they do not actually detail the contents of this “toolkit”.

As part of this toolkit the IOM feel that “the development of clinical questionnaire or history tools that are valid across populations of patients should be an urgent priority”. Toward this end Appendix D of their report comprises “Questionnaires and Tools That May Be Useful for Assessing ME/CFS Symptoms” with the footnote that “Further evidence of utility is needed”.

Our main aim was to sensitize the physicians (and in particular hematologists) that PPBL was not only a diagnosis emerging from the lab, but that it could be initialy suspected for very tired smoking women. Our aim was not to perform a thorough description of a border diagnosis of SEID expertized by an internist. We apologize if the method had some flaws and to your point of view a lack of rigor. Out take home message was that PPBL patients did realy not feel safe.

Nevertheless, we were aware of the controversies about CFS and SEID, thus we asked before the survey for the opinion and advice of specialists – in this context, the board of the French Association of Chronic Fatigue Syndrome – to validate our approach and the tools used. And we refer patients as soon as possible to a SEID specialist (Pr JD DeKorwin) to assess our “non-specialist” opinions (sorry, all authors but JDDK are hematologists !).

We believe that we have taken the maximum precautions to validate our clinical data, and a maximum of precautions to distribute a relevant questionnaire. Nevertheless, we thank the reviewer for its warming about and include his relevant point of view in our manuscript. We added a whole paragraph mentioning that while the IOM do recommend the development of “a toolkit appropriate for screening and diagnosing patients with ME/CFS”, they do not actually detail the contents of this “toolkit” and that our protocol was not indeed fully validated.

In my original review I mentioned a study by Asprusten et al (2018) which follows the IOM recommendations when applying SEID diagnostic criteria to an adolescent CFS cohort. This study used variables from a total of 8 validated questionnaires to operationalise the SEID criteria, and then used baseline data to decide whether a patient fulfilled those criteria or not. Two CFS subgroups (SEID vs Non SEID) were compared across baseline characteristics, as well as a wide range of cardiovascular, inflammatory, infectious, neuroendocrine and cognitive variables. I find no evidence to show that this present research or the original Campagne (2016) study employed a similar level of scientific rigor when developing their own SEID questionnaire.
 Aspruten et als. findings question the discriminant and prognostic validity of the SEID diagnostic criteria in adolescent CFS. In their response to my comments, the present authors acknowledge “the controversy arising from the IOM/NAM's proposals to better define ME/CFS under the SEID label”. However, they fail to discuss the implications that a lack of discriminant and prognostic validity of the SEID diagnostic criteria might have for their use in both research and clinical practice.
 The Aspruten study suggests that the SEID criteria tend to select patients with depressive symptoms. The present study does recognize that “skepticism about the role of psychological factors in ME/CFS and high rates of depression in patients with PPBL can lead to misdiagnosis of ME/CFS as a psychiatric disorder”. However, there were no controls for depression employed in the present study, nor is it discussed as a potentially confounding factor.

We thank the reviewer for highlighting Aspruten’s paper. We added it in our reference list and add a few comment in our text to warm readers to the persistent controversies for a rigourous SEID investigation. However, it is not possible at this time to to resume our study by recreating a specific questionnaire. It is even possible that patients who consult for unexpected lymphocyte abnormalities do not fully understand the meaning of it. On the other hand, we have planned to check the hematological anomalies of a series of patients with SEID, and we will then take into account your suggestions.

I was not aware of PPLB before reading this paper and thank the authors for drawing my attention to this rare condition.

            We thank reviewer 3 for his encouraging opinion. Making readers aware of this poorly-known pathology was one of our goals.

            There is certainly potential benefit from further exploration of a possible link with ME/CFS. Is it a trigger for ME/CFS, a comorbidity, or a condition with symptoms that resemble ME/CFS? The possible link with EBV is certainly interesting and, of course, other possible viral etiologies such as COVID 19. PPLB should likely be considered for differential diagnosis when evaluating patients for possible ME/CFS. However, I do not believe this present study adds any clarity to the current situation regarding diagnosis and/or management of either ME/CFS or PPLB.

            We thank the expert reviewer about ME/CSF for his comments. We agree that our description of the questionnaire was indeed incomplete. However, we did our best, and would be to much hard work to start over this study.

Reviewer 2 Report

After revision, the manuscript is ok

Author Response

The authors thank reviewer 2

English has been checked again

Reviewer 3 Report

I think the revised manuscript has significantly improved with responding reviewers' comments.

Author Response

The authors thank reviewer 3

This manuscript is a resubmission of an earlier submission. The following is a list of the peer review reports and author responses from that submission.

Round 1

Reviewer 1 Report

While I am not familiar with PPBL, I have considerable experience with ME/CFS. Adding to the body of knowledge for poorly understood conditions is commendable but I do not feel that the current  study really does much in this vein. If the intent is to diagnose ME/CFS (SEID) in patients previously diagnosed with PPBL, then there are serious flaws in the methodology and instrumentation.

The paper indicates using “the IOM tool kit  questionnaire, translated and validated by the French CFS association (ASFC)” (p.4). The IOM (now called National Academy of Medicine, NAM) did propose diagnostic criteria  and a diagnostic algorithm for ME/CFS along with a recommendation to rename ME/CFS as SEID. I am not aware of a specific SEID questionnaire being developed by the IOM.

Careful reading of both this current paper and the referenced Campagne et al (2016) paper, my conclusion is that the French language questionnaire used in both instances was constructed using the proposed IOM SEID diagnostic criteria. It is not therefore a translation of an existing questionnaire but a newly constructed instrument. As such, there are no previously published validity and reliability data or psychometric properties that would support its use in a research context. The validation process for the newly created French SEID questionnaire described in Campagne et al. does not meet accepted criteria for the development of a valid and reliable research  instrument. It comprised merely correlating self-reported symptoms with self-reported diagnoses for a self-selected sample of members from the French Association of Chronic Fatigue Syndrome (ASFC).

While it might be useful to highlight similarities and differences between symptoms experienced by patients diagnosed with PPBL and those with an ME/CFS diagnosis, the proposed IOM  SEID diagnostic criteria are considered controversial and not widely accepted by the ME/CFS research community (e.g., Jason et al., 2015, 2017, Asprusten, 2018). The discriminant and prognostic validity of the SEID diagnostic criteria has been questioned on the grounds that they are overly inclusive and prone to misdiagnose individuals who have other conditions. In an article  cited in this current paper, the author notes “ important methodological shortcomings” in the new SEID diagnostic criteria (Twisk, 2015). If the proposed SEID diagnostic criteria have serious shortcomings when it comes to specificity for ME/CFS, their use in research with other conditions must be questioned.

The current paper does address some of these issues regarding specificity but fails to convince me that using the IOM SEID criteria to better describe/understand PPBL was either legitimate or worthwhile. In finding that a significant number of persons with PPBL met criteria for diagnosis of SEID, it supports criticisms that the IOM criteria are over inclusive and therefore unsuitable for research purposes.

For discussion of SEID criteria and further references see:

Leonard A. Jason, Madison Sunnquist, Kristen Gleason & Pamela Fox (2017) Mistaken conclusions about systemic exercise intolerance disease being comparable to research case definitions of CFS: A rebuttal to Chu et al., Fatigue: Biomedicine, Health & Behavior, 5:4, 231-238, DOI: 10.1080/21641846.2017.1362780

Reviewer 2 Report

This is an interesting manuscript analyzing in details the symptoms associated with PPBL in a relatively large series of patients included in the French National Registry. The research was conducted using an anonymous questionnaire sent by post to 50 patients included in the registry. Aim of the research is to better understand the characteristics of the fatigue associated with PPBL and to study its relationship with systemic exertion intolerance disease (SEID). The Authors conclude that many patients with PPBL presented with symptoms suggestive of SEID, and then medical staff should be encouraged to better assess this peculiar condition.

COMMENTS

 I consider it is essential to include the strengths and weaknesses of a postal survey in the text. In particular, the limits of this methodology. I think it might be useful to add some sentences about the limits of a postal survey in the Discussion paragraph

Discussion. "This survey highlighted that nearly three-quarter of the patients with PPBL met the SEID criteria."

  Attention: you sent the questionnaire to 50 patients and only 32 replied. Of these, 23 (46% of all patients in the database) met the criteria for SEID diagnosis. I therefore think it is more prudent to tone down the statement

Your research is clearly aimed at symptom assessment. However, "50 among them were exhaustively investigated and regularly followed up (clinical evaluations, full blood count, lymphocyte immunophenotyping, HLA investigation, serum immunoglobulins measurement, genetics, etc.)." I believe a brief summary or an implementation of Tab 1 with more exhaustive reporting of the biological data from your patient series could improve the manuscript. For instance: How many patients had binucleated lymphocytes?  Kappa and lambda light-chain expression?  Rearrangement of immunoglobulin heavy chain genes? Secondary neoplasia? Evolution in NHL?    Etc..

I agree with the authors that many studies deal with morphological and biological aspects in general, while little attention has been given to symptoms. However, E Cornet, J F Lesesve, H Mossafa, G Sébahoun, reporting the long-term follow-up of 111 patients stated that they were either asymptomatic or had minor and nonspecific complaints, such as fatigue. Even in my  personal experience, I have not noticed a relevant symptomatology. How do you explain these notable differences with your experience? Could there be an exaggeration of symptoms by patients linked to the modality of collection of symptoms through a questionnaire? or bias in the patient selection ?

Reviewer 3 Report

This study revealed that majority of persistent polyclonal B-cell lymphocytosis(PPBL) patients actually met criteria of SEID, whose pathogenesis is not resolved and constituted an unmet medical need. This report is precious in that it would be the first one describing the relationship between PPBL and SEID. The study used registered thoroughly investigated PPBL patients, which added more solid evidence.

The reviewer think the value of this study can be increased by adding a few more analyses.

1. The relation between PPBL clinical information (such as age, duration, severity - if there is, comorbidities) and each symptoms in the criteria of SEID. In the same token, relation between PPBL clinical information and frequency and severity of SEID symptoms would be an important information.

2. What was the correlation among each SEID symptom?
The authors could make a correlation matrix of each symptom, which may help understand characteristics of SEID in PPBL patients.

3. Sub-category analysis.
Inflammatory/autoimmune comorbidities need special attention, because these diseases often cause SEID-like symptoms. How about to showing sub-analysis data - PPBL without any inflammatory/autoimmune comorbidities and PPBL with such comorbidities?  Another potentially important comorbidities are psychiatric illnesses such as depression or PTSD in the same reason.

4. Representative case presentation.
The readers could understand the actuality of PPBL patients, if the paper included one or two representative cases. Moreover, chronological order of  PPBL diagnosis and the onset of SEID symptoms is interesting to know. 

5. Data presentation.
The reviewer advises to make some graphs to show the frequency of each symptoms. (adding data of clinical information/SEID symptoms/severity relation)

7. Terminology
The reviewer noticed some unconventional terminology.
Line 17: polyalgic
Line 199: dysimmunity

Table 1: Auto-immunity (%)